# Knowledge Corpus Error in Question Answering

**Yejoon Lee**[*]
Seoul National University
leeyejoon@snu.ac.kr

**Philhoon Oh**
KAIST AI
philhoonoh@kaist.ac.kr

**James Thorne**
KAIST AI
thorne@kaist.ac.kr

## Abstract

Recent works in open-domain question answering (QA) have explored generating context passages from large language models (LLMs), replacing the traditional retrieval step in the QA pipeline. However, it is not well understood why generated passages can be more effective than retrieved ones. This study revisits the conventional formulation of QA and introduces the concept of *knowledge corpus error*. This error arises when the knowledge corpus used for retrieval is only a subset of the entire string space, potentially excluding more helpful passages that exist outside the corpus. LLMs may mitigate this shortcoming by generating passages in a larger space. We come up with an experiment of paraphrasing human-annotated gold context using LLMs to observe knowledge corpus error empirically. Our results across three QA benchmarks reveal an increased performance (10% - 13%) when using paraphrased passage, indicating a signal for the existence of knowledge corpus error.[1]

## 1 Introduction

Large language models (LLMs) generate surprisingly fluent and informative texts. This led to many works utilizing the text data generated by these models for purposes such as instruction tuning (Honovich et al., 2022; Wang et al., 2022) and improving reasoning capability (Zelikman et al., 2022; Ho et al., 2023; Magister et al., 2023).

Open-domain question answering (QA) (Chen et al., 2017) is a task where retrieving relevant passages from a corpus of factual information such as Wikipedia is standard practice. Recent works have attempted to generate such passages from LLMs, replacing the retrieval step of the traditional pipeline (Sun et al., 2023; Yu et al., 2023). Despite their success, it is not well understood why these generated passages could be more effective than

---
[*] Work done during internship at KAIST AI

[1] Our code is available at this repository

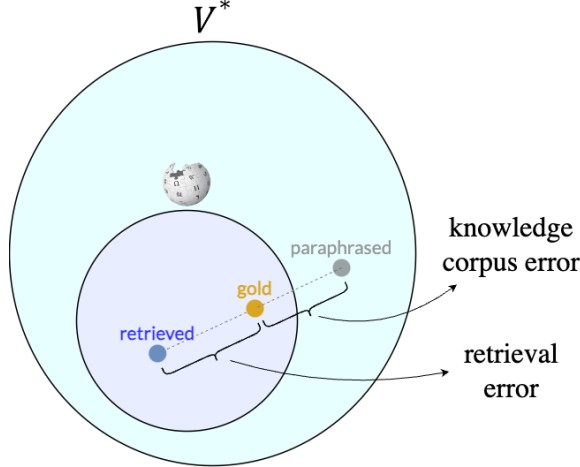

Figure 1: Illustration of our framework. Each dot represents a passage. *Retrieval error* refers to an error from failing to retrieve the gold passage. *Knowledge corpus error* refers to an error from discarding better passages outside the knowledge corpus, e.g., Wikipedia, which is inevitable in any retrieval setting. See §3 for details.

retrieved passages. These recent advancements lack robust links to prior research in QA posing a challenge to a holistic understanding.

By revisiting the formulation of answer generation with retrieved passages (Guu et al., 2020; Lewis et al., 2020; Singh et al., 2021), we identify the then-overlooked gap, which has become significant now due to the advance in LLMs (Brown et al., 2020). Our discussion starts with the observation that the *knowledge corpus* from which the passages are retrieved is only a subset of the possible string space. More helpful passages to the reader may exist outside the knowledge corpus. Unfortunately, retrieval, by definition, cannot utilize passages outside the knowledge corpus, potentially causing a shortfall. We refer to this as *knowledge corpus error*. In contrast, LLMs can generate passages from the entire string space, which may mitigate the inherent limits of retrieval.

We empirically demonstrate the presence of

knowledge corpus error where a passage from outside of Wikipedia outperforms the human-annotated gold passage inside Wikipedia in question answerin. We design an experiment of paraphrasing human-annotated gold context with LLMs. Experiments with four QA benchmarks, NQ (Kwiatkowski et al., 2019), HotPotQA (Yang et al., 2018), StrategyQA (Geva et al., 2021), and QASC (Khot et al., 2019) result in 10% - 13% gain in reader performance across three benchmarks, when using paraphrased passages. This gain supports our hypothesis that there exist more helpful passages than the gold passage outside the knowledge corpus.

## 2 Related Work

### 2.1 Leveraging LLM-generated text

As the quality of text generated from LLMs has improved through larger models (Kaplan et al., 2020) and instruction tuning (Sanh et al., 2022; Wei et al., 2022a), many works have sought to use these models as data sources in other NLP tasks. Text generated from LLMs has been used for generating datasets for instruction finetuning (Honovich et al., 2022; Wang et al., 2022), improving reasoning (Zelikman et al., 2022; Ho et al., 2023; Magister et al., 2023), and many other purposes (Liu et al., 2022a; Ye et al., 2022; Haluptzok et al., 2023).

Recently, there has been growing attention towards open-source LLMs (Touvron et al., 2023) finetuned on instructions generated from proprietary LLMs, such as Alpaca (Taori et al., 2023), Koala (Geng et al., 2023) and Vicuna (Chiang et al., 2023). Text generated from these models purportedly match quality of those from proprietary LLMs (Chiang et al., 2023), but this assertion remains disputed (Gudibande et al., 2023). Understanding the role of LLM-generated text will serve as an important aspect of this discourse.

### 2.2 Knowledge-intensive NLP and retrieval

Knowledge-intensive NLP, such as open-domain QA (Chen et al., 2017) and fact verification (Thorne et al., 2018), requires substantial factual knowledge that may change over time. Therefore, these tasks were originally envisioned to incorporate the retrieval of relevant passages from the knowledge corpus (Chen et al., 2017). In the typical retrieve-then-read pipeline (Karpukhin et al. 2020, *inter alia*), a pipeline of models, first selects $k$ passages from a retrieval function which are then used to condition answer generation from reader (Izacard and Grave 2021, *inter alia*).

Meanwhile, the success of pre-trained language models (Raffel et al., 2020) and the associative memory properties learned during training (Petroni et al., 2019) has allowed researchers to revisit closed-book QA, in which models answer the questions without being provided a passage. Closed-book QA was demonstrated to be effective both in in-context learning (Brown et al., 2020) and supervised learning (Roberts et al., 2020).

Recent works on chain-of-thought prompting has shown that generating intermediate steps before giving an answer improves the reasoning capability of LLMs (Wei et al., 2022b; Wang et al., 2023). Inspired by this, recent works prompt LLMs to generate the intermediate step of QA, which is the passages (Sun et al., 2023; Yu et al., 2023). These passages are subsequently fed into the reader, either supervised FiD (Izacard and Grave, 2021) or in-context learning LLMs (Brown et al., 2020). Despite their success, these methods require a very large scale and risk generated passages containing stale or non-factual information. Moreover, it is not fully explained why generating passages may have advantages over retrieval.

## 3 Analytic Discussion

Our task formulation follows retrieval augmented models for QA (Guu et al., 2020; Lewis et al., 2020; Singh et al., 2021). These works view contexts as a latent variable for the QA model (Lee et al., 2019).

### 3.1 Setup

Let $V^*$ be the infinite set of all possible strings over vocabulary tokens in $V$, including the empty string. An instance of a QA dataset consists of a triple $(q, a, c)$: question $q$, answer $a$, and context $c$, where $q, a, c \in V^*$. Typically, the context $c$ is retrieved from the knowledge corpus $\mathcal{Z}$, such as Wikipedia, where $\mathcal{Z} \subset V^*$.

### 3.2 QA Task Formulation

The goal of QA is to learn a distribution $p(a|q)$, where models decode a string $a$ that acts as an abstractive answer to the query (Lewis et al., 2020; Izacard and Grave, 2021). One can directly prompt a language model to obtain an answer $a$, given question $q$ (where context $c$ is implicitly the empty string), relying only on model parameters in closed-book QA (Roberts et al., 2020; Brown et al., 2020).

$$\hat{a} = \arg\max_{a \in V^*} p(a|q) \quad (1)$$

However, direct prompting is often difficult to learn and barely discloses its inner working. Therefore, a popular approach is to marginalize $p(a|q)$ over contexts in the knowledge corpus (Guu et al., 2020; Lewis et al., 2020; Singh et al., 2021). As it is intractable to calculate the probability for all the contexts in the knowledge corpus, $p(a|q)$ is approximated to the sum of probabilities for top $k$ contexts from $\mathcal{Z}$. $Topk(\mathcal{Z}, q)$ denotes the set of resulting top $k$ passages after the retrieval with a query $q$.

$$p(a|q) \approx \sum_{c \in Topk(\mathcal{Z}, q)} p(a|q, c)p(c|q) \quad (2)$$

The gap in this formulation is that relevant context $c$ may exist outside of the knowledge corpus $\mathcal{Z}$. This makes the sum of marginal probabilities over $\mathcal{Z}$ only an approximation. The true probability would require the summation of marginal probabilities over the infinite string space $V^*$.

$$\begin{aligned} p(a|q) &= \sum_{c \in S} p(a|q, c)p(c|q) \\ &\approx \sum_{c \in \mathcal{Z}} p(a|q, c)p(c|q) \\ &\approx \sum_{c \in Topk(\mathcal{Z}, q)} p(a|q, c)p(c|q) \end{aligned} \quad (3)$$

### 3.3 Knowledge corpus error

Equation 3 details two steps of approximation, which results in two sources of potential error in QA using contexts. The first source of error is introduced when the entire knowledge corpus $\mathcal{Z}$ is approximated to top $k$ retrieved contexts, $Topk(\mathcal{Z}, q)$. This error, which we denote ***retrieval error***, can be mitigated by better retrieval methods or increasing $k$, the number of contexts. On the other hand, the second source of error is introduced when the entire string space $V^*$ is approximated to knowledge corpus $\mathcal{Z}$. This error is rooted in the use of knowledge corpus itself, hence we denote it as ***knowledge corpus error***. To elaborate, for some $\tilde{c} \notin \mathcal{Z}, p(\tilde{c}|q) > p(c \in \mathcal{Z}|q)$, but $p_{retriver}(\tilde{c}|q) = 0$ whereas $p_{LLM}(\tilde{c}|q) > 0$.

For a query $q$, $p(c|q)$ is sufficiently small for most contexts $c$. This allows these terms be ignored by setting $p(c|q)$ to zero. For instance, top-$k$ retrieval is essentially setting $p(c|q)$ to zero

for $c \notin Topk(\mathcal{Z}, q)$. For contexts outside the knowledge corpus, $\tilde{c} \notin \mathcal{Z}$, applying Bayes' rule, $p(\tilde{c}|q) \propto p(q|\tilde{c})p(\tilde{c})$, where the retrieval-based task formulation is setting the prior $p(\tilde{c}) = 0$. Knowledge corpus error may explain why reader models can benefit from generated contexts (Sun et al., 2023; Yu et al., 2023) as LLMs can generate strings from the set $V^* \supset \mathcal{Z}$.

## 4 Empirical Observation

To observe knowledge corpus error, we study the effect of *paraphrasing* human-annotated *gold* contexts from QA dataset. Gold context $c_{gold} \in \mathcal{Z}$ is what humans annotated as the supporting passage for a given question-answer pair. While human annotation may be imperfect, we assume that this $c_{gold}$ acts as the best available passage from the knowledge corpus $\mathcal{Z}$, i.e., there is no retrieval error. In our experiment, $c_{gold}$ is paraphrased into $c_{paraph}$, by prompting LLMs with $c_{gold}$ and $q$. Then, $c_{gold}$ and $c_{paraph}$ are separately fed into the reader to compare the performance. As $c_{gold}$ is the best available context, any gains from paraphrasing should be attributed to reduced knowledge corpus error.

### 4.1 Experimental setup

For a single instance of a QA dataset $(q, c_{gold}, a)$ and a paraphrased context $c_{paraph} = Paraph(c_{gold}, q)$, we compare model performance in two settings without and with paraphrasing: $Read(q, c_{gold})$ and $Read(q, c_{paraph})$. Both $Paraph()$ and $Read()$ are function calls to LLMs, GPT-3.5 (gpt-3.5-turbo[2]) and Claude (claude-1[3]). Experiments were conducted in June 2023.

### 4.2 Benchmarks

For benchmarks, we used NQ (Kwiatkowski et al., 2019), HotPotQA (Yang et al., 2018), StrategyQA (Geva et al., 2021), and QASC (Khot et al., 2019). NQ consists of factual questions which can be answered with a single passage. Unlike NQ, Hot-PotQA consists of questions that require reasoning across multiple passages, known as multi-hop QA. StrategyQA and QASC further extend this multi-hop setting by requiring more implicit reasoning.

For gold context $c_{gold}$, we use the paragraph(s) from Wikipedia, which are part of the annotations in NQ, HotPotQA, and StrategyQA. For QASC,

---

[2]https://api.openai.com/v1/chat/completions
[3]https://api.anthropic.com/v1/complete

| Benchmarks | Reader: GPT | | | Reader: Claude | | | Average gap between |
|---|---|---|---|---|---|---|---|
| | Gold | GPT | Claude | Gold | GPT | Claude | gold and paraphrased |
| NQ exact match (%) | 40.9 | 39.9 | 44.3 | 18.3 | 21.3 | 35.5 | 3.125 |
| HotPotQA exact match (%) | 36.3 | 38.6 | 43.4 | 47.6 | 50.9 | 54.2 | 4.825 |
| StrategyQA accuracy (%) | 54.6 | 56.4 | 70.5 | 68.9 | 75.5 | 76.5 | 7.975 |
| QASC accuracy (%) | 95.7 | 92.4 | 91.1 | 86.3 | 75.7 | 76.9 | - 6.975 |

Table 1: Performance of each reader when given original gold context ("Gold"), paraphrased context with GPT ("GPT"), and paraphrased context with Claude ("Claude"). Red indicates an increase in performance after paraphrasing, implying knowledge corpus error has been observed. Blue indicates a decrease in performance after paraphrasing, implying knowledge corpus error has not been observed.

where such paragraph does not exist, we treat the seed facts that were used to create questions as gold context. In multi-hop QA, we concatenate all the contexts into a single context. See Appendix B.1 for details.

### 4.3 Results

We report the results in Table 1. Paraphrased context outperforms the original gold context for most cases in NQ, HotpotQA, and StrategyQA. This means that paraphrased passages were more helpful than the gold passages, implying the existence of knowledge corpus error. Moreover, using the context paraphrased by different model did not cause any performance depredation, indicating some level of universality in the helpfulness of the passages. We provide further analysis of this finding in Appendix E.

**QASC**. We attribute the degradation in QASC for two reasons. First, the seed facts, which we considered as gold contexts in QASC, are not from a raw corpus. The seed facts are manually selected from cleaned knowledge sources like WorldTree corpus. This is problematic as the gold contexts we are using represent the best-case scenario in retrieval, thereby eliminating any retrieval error. Second, distractor options in multiple-choice question confuses the model to generate a passage relevant to those options. This results in a passage containing distracting information for answering the question. Examples in Table 8 illustrate these two points well.

### 4.4 Qualitative Analysis

After manually examining a sample of results from the empirical study, we identify three common factors contributing to knowledge corpus error.

**1. Increased focus on the question** Gold passages are a very small subset of facts from Wikipedia, with the communicative intent to generally inform about the subject. Therefore, gold passages inevitably include information that is irrelevant to the question. LLMs can only filter the helpful information from the gold passage during the paraphrasing. In fact, it has been shown that when both retrieved and generated passages contain the correct answer, the FiD reader can produce more correct answers when reading the generated passages (Yu et al., 2023). And furthermore, models are sensitive to related but irrelevant information in a phenomena called *damaging retrieval* (Sauchuk et al., 2022; Oh and Thorne, 2023). Query-focused paraphrasing acts as an information filter mitigating damaging retrieval.

**2. Chain-of-thought** Some questions require a composition of facts to answer the question. In such case, we observe that paraphrasing is acting in a manner similar to chain-of-thought (Wei et al., 2022b). This also highlights the inherent limit of corpus such as Wikipedia, where the explicit composition of information is seldom given.

In the second example of Table 2, the question requires combining two distinct facts, one about the military unit (VMAQT-1) and another about Irish mythology (Banshee). The paraphrased context acts somewhat akin to chain-of-thought, resulting in a more helpful context.

**3. Incorporation of commonsense knowledge** Commonsense knowledge plays a crucial role in understanding the world (Davis and Marcus, 2015), but not often explicitly stated, especially in a corpus such as Wikipedia. Language models are known to possess a degree of tacit knowledge (Petroni et al., 2019), which can be utilized by knowledge generation (Liu et al., 2022b). We observe that during paraphrasing, commonsense knowledge is elicited, aiding the reader.

The third example of Table 2 illustrates how com-

| Source | Question / Answer | Gold | Paraphrased |
|---|---|---|---|
| Increased focus on the question | Q. who plays charles on have and have nots
A. Nick Sager | Title: The Haves and the Have Nots (TV series)
The Road to Hell", it was revealed that David had a pulse while Erica's fate was a casualty and died from the explosion. Her death was the fuel for David's warpath of vengeance on Veronica. - **Charles Frederickson (Nick Sager)**: (Season 4–present) The president-elect and the love interest to Candace, whom she uses to help her get her money back. (...) | **Charles, played by Nick Sager**, is one of the main characters on the show The Haves and the Have Nots. He is portrayed as the president-elect and love interest of the character Candace. (...) |
| Chain-of-thought | Q. The VMAQT-1 logo is a female spirit in Irish mythology who heralds the death of what?
A. a family member | Title: VMAQT-1
(...) The **VMAQT-1 logo is the Banshee**, an Irish mythological figure foretelling death. Its motto is "Tairngreacht Bas," Gaelic for "Death Foretold."
Title: Banshee
A banshee ( ; Modern Irish "bean sí", from Old Irish: "ban síde" , ] , "woman of the fairy mound" or "fairy woman") is a female spirit in Irish mythology who heralds the **death of a family member**, usually by shrieking or keening. (...) | **The VMAQT-1 logo features a banshee**, which is a female spirit in Irish mythology. According to Irish mythology, banshees shriek or keen to foretell the **death of a family member**. (...) |
| Incorporation of commonsense knowledge | Q. Was Bruce Lee absent from the 1964 University of Washington graduation ceremony?
A. True | Title: Bruce Lee
Lee is best known as a martial artist, but he also studied drama and Asian and Western philosophy while a student at the University of Washington and throughout his life. (...)
Title: Bruce Lee
Lee **dropped out of college in early 1964** and moved to Oakland to live with James Yimm Lee. (...) | There is no mention of him attending or participating in the graduation in either document. The documents instead mention that in early 1964, Lee dropped out of college and moved to Oakland (...) While this evidence is not definitive, it suggests that **Lee probably did not attend** or participate in the University of Washington's 1964 graduation ceremony. |

Table 2: Qualitative analysis on the source of knowledge corpus error. Examples are respectively from NQ, HotPotQA, and StrategyQA (from top to bottom). All paraphrases are done by Claude. The **bold** part highlights how paraphrased context may be more helpful than gold context. See 4.4 for detailed explanation.

monsense knowledge — *someone who dropped out of college will probably not attend graduation ceremony* — is induced during paraphrasing.

## 5 Conclusion

In this work, we demonstrate that generated contexts may be more helpful than retrieved contexts in open-domain question answering. By revisiting the formulation of question answering, we identify a gap where retriever inevitably ignores potentially helpful contexts outside of the corpus. We call this *knowledge corpus error*, and design an experiment in order to observe knowledge corpus error empirically. Paraphrasing the human-annotated gold contexts with LLMs led to increased reader performance in 3 out of 4 QA benchmarks, implying the existence of knowledge corpus error.

## Acknowledgments

This work was supported by Institute of Information & communications Technology Planning & Evaluation (IITP) grant funded by the Korea government (MSIT) (No.2019-0-00075, Artificial Intelligence Graduate School Program (KAIST)) and Artificial intelligence industrial convergence cluster development project funded by the Ministry of Science and ICT(MSIT, Korea) & Gwangju Metropolitan City.

## Limitations

The first limitation of this work is that it did not employ it in a retrieval setting. We used a gold context, which we assume is the best-case scenario in retrieval. However, retrieved contexts in real retrieval setting (Karpukhin et al., 2020), i.e., contexts in knowledge corpus other than the gold context, may deviate significantly from the gold context. Therefore, it is hard to discuss the effect of paraphras-

ing and the degree of knowledge corpus within retrieval.

The second limitation of this work is that it did not address the practical way to marry retrieval and generation via LLMs. Regardless of the seeming benefit of context generation, this approach suffers from issues such as information staleness and hallucination. Contemporaneous works explore various methods to leverage benefits of both retrieval and generation (He et al., 2022; Jiang et al., 2023; Xu et al., 2023). This work is primarily concerned with the analytic understanding of how generation may have advantages over retrieval. We believe our work can inspire future contributions on empirical method of incorporating retrieval and generation.

The third limitation of this work is that its scope was limited to question answering. Conditioning generation on retrieved context is a well-studied approach in language modeling (Khandelwal et al., 2020; Yogatama et al., 2021; Borgeaud et al., 2022). It will be worth exploring how knowledge corpus error manifests within language modeling.

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

# A Dataset

**NQ**: We use the KILT (Petroni et al., 2021) version of NQ[4]. We use the dev split after excluding the instances where context does not include answers, which results in 2532 samples.

**HotPotQA**: We use the dataset from its original

---

[4]https://github.com/facebookresearch/KILT

source[5]. We use the dev split, which includes 7405 samples. To reduce the inference cost, we only use the subset of first 1531 samples.

**StrategyQA**: We use the dataset from its original source [6]. We use the training split, which includes 2290 samples. We use the training split because the dev split contained too few (490) examples.

**QASC**: We use the dataset from its original source [7]. We use the dev split, which includes 926 samples.

## B Detailed experimental setup

### B.1 Selection of gold passage

**NQ**: NQ contains a set of provenances for possible answer contexts. For the experiments, we select the gold passages from the provenances that include at least one of the candidate answers. When there are multiple good passages, we employ the very first one.

**HotPotQA**: HotPotQA contains 2 gold paragraphs from Wikipedia for each question. A gold passage is simply the concatenation of these two. Note that we do not utilize fine-grained sentence-level annotation in 2 paragraphs.

**StrategyQA**: StrategyQA contains decomposition steps to solve the question. Each of these steps may be attached with a supporting paragraph from Wikipedia. A gold passage is the concatenation of all these paragraphs throughout the whole steps. Among three different annotated decomposing steps in the dataset, we use the first one.

**QASC**: QASC contains two facts that are combined to create a question. These facts are selected from a cleaned knowledge source. A gold passage is simply the concatenation of these two facts.

The title of the passage is prepended to the passage in cases where titles are available (NQ, HotPotQA, and StrategyQA).

### B.2 Details on generation

**NQ**: During reading, we used 3-shot prompting, where the 3-shot demonstrations are sampled from GPT-3.5 with questions from the dev split of NQ. Note that these questions are excluded from the experiment. Max tokens to generate was set to 500 in paraphrase and 25 in read.

**HotPotQA**: Max tokens to generate was set to 300 in paraphrase and 10 in read.

**StrategyQA**: Max tokens to generate was set to 300 in paraphrase and 10 in read.

**QASC**: Max tokens to generate was set to 100 in paraphrase and 10 in read. Temperature during generation was set to 0.8 in paraphrase and 0.4 in read.

We used 3-shot prompting for reading in NQ but otherwise used zero-shot prompting. Other generation keyword arguments are set to default if not specified. For the prompts used, see Table 4.

Hyperparameters related to generation are decided mainly through trial-and-error. For example, max tokens was adjusted according to few preliminary samples. We tried to tweak temperature for QASC after observing deviant result, but only had minor impact. 3-shot setup was chosen for NQ because the performance was too low in zero-shot.

## C Details on evaluation

Evaluating exact match or accuracy may be non-trivial in a generative setting. Hence, we follow the previous works (Kojima et al., 2022; Yu et al., 2023).

**NQ (Exact Match)**: Following Yu et al. 2023, we measure exact match of the output string after normalization.

**HotPotQA (Exact Match)**: Similarly as Yu et al. 2023, we measure exact match of the output string after normalization.

**StrategyQA (Accuracy)**: Following Kojima et al. 2022, we measure accuracy by picking up the first "yes" or "no" encountered in the text after removing unnecessary letters.

**QASC (Accuracy)**: Similarly as Kojima et al. 2022, we measure accuracy by picking up the first large letter out of A to H encountered in the text.

## D Inference Cost

We used OpenAI and Anthropic's API to use their LLMs. The cost for OpenAI's API is estimated to be around $40 to $50. Anthropic's API has not cost any as we were on the free version.

## E Accordance between heterogeneous reader models

For a majority of the examples, two readers accord with each other, i.e., both are correct or wrong, and this ratio is even higher in paraphrased contexts.

---

[5]https://hotpotqa.github.io/
[6]https://allenai.org/data/strategyqa
[7]https://allenai.org/data/qasc

| Given context | Both | | Only correct for | |
|---|---|---|---|---|
| | correct | wrong | GPT | Claude |
| Gold | 27 | 43 | 9 | 21 |
| GPT-paraph | 33 | 44 | 5 | 18 |
| Claude-paraph | 39 | 42 | 4 | 15 |

Table 3: Accordance between two heterogeneous reader models, GPT and Claude. The table is based on the reader performance (exact match (%)) in HotPotQA when given gold context, GPT-paraphrased context, and Claude-paraphrased context (from top to bottom).

## F    Examples

For examples of paraphrased context, see Table 5 through 8.

| Benchmark | Paraphrase | Read |
|---|---|---|
| NQ | Paraphrase a background document in your own words to answer the given question.

Question: {q}

Document: {c} | Referring to the passage, find the correct answer (just one entity) to the given question. I will first show you few examples.

Passage: {c1}
Question: {q1}
Answer: {a1}

*example continues*

Passage: {c}
Question: {q}
Answer: |
| HotPotQA | Paraphrase the background documents into a single document in your own words to answer the given question.

Question: {q}

Documents: {c} | Referring to the passage, find the correct answer (just one entity) to the given question.

{c}

{q} |
| StrategyQA | Paraphrase the background documents into a single document in your own words to answer the given question.

Question: {q}

Documents: {c} | Read the passage and answer the question with yes or no.

{c}

{q} |
| QASC | Paraphrase the background documents into a single document in your own words to answer the given question.

Question: {q}

Documents: {c} | Read the passage and answer the question with one of A, B, C, D, E, F, G, or H.

{c}

{q} |

Table 4: **Prompts used for paraphrasing and reading.** {q}, {c}, and {a} each indicate question, context, and answer from the test instance. {q1}, {c1}, and {a1} indicate one of the demonstrations used for few-shot prompting. Note that the full few-shot prompt format is not displayed for conciseness.

| Benchmark | Gold | Paraphrased |
|---|---|---|
| NQ | **Question: what does the pearl symbolize in the bible**

Title: Parable of the Pearl

Parable of the Pearl The Parable of the Pearl (also called the Pearl of Great Price) is one of the parables of Jesus. It appears in and illustrates the great value of ==the Kingdom of Heaven==. It is the penultimate parable in Matthew 13, just before the Parable of the Dragnet. It immediately follows the Parable of the Hidden Treasure, which has a similar theme. It does not appear in the other synoptic gospels but a version of this parable does appear in the non-canonical Gospel of Thomas | **GPT:** The Pearl of Great Price is a parable told by Jesus that is found in the book of Matthew. It teaches us about the immense worth of ==the Kingdom of Heaven==. This parable is situated just before the Parable of the Dragnet and comes after the Parable of the Hidden Treasure, which has a similar meaning. (...)
*GPT: X -> X, Claude: X -> O*

**Claude:** In summary, the pearl symbolizes ==the Kingdom of Heaven== in the Bible. This is illustrated in the Parable of the Pearl told by Jesus, where a merchant finds a pearl of great value and sells all his possessions so that he can buy it. The pearl represents ==the Kingdom of Heaven==, which is so valuable that one should sacrifice everything else to obtain it.
*GPT: X -> O, Claude: X -> X* |
|  | **Question: when did the song here comes the boom come out**

Title: Boom (P.O.D. song)

Boom (P.O.D. song) "Boom" is a song by American rock band P.O.D. It was released in ==May 2002== as the third single from their second major label studio album "Satellite". While it did not chart as well as the albumś previous singles, the song has appeared significantly in film and television. "Boom (The Crystal Method remix)" was included on the remix album "Community Service" and as a bonus track on the special edition re-release of "Satellite" available August 27 | **GPT:** "Here Comes the Boom" is a song by P.O.D., an American rock band. It was released in ==May 2002== as the third single from their album "Satellite." Although it didn't perform as well as the previous singles from the album, it has been featured in numerous movies and TV shows. (...)
*GPT: X -> O, Claude: X -> O*

**Claude:** The song 'Here Comes the Boom' by the rock band P.O.D. was released in ==May 2002==. It was the third single from their album 'Satellite' released that year. Though the song did not chart highly, it has been featured in various films and TV shows. A remixed version of the song by The Crystal Method was included in P.O.D.'s remix album 'Community Service' as well as in a special re-release of the 'Satellite' album in August 2002.
*GPT: X -> O, Claude: X -> X* |

Table 5: **Examples of gold contexts and paraphrased contexts in NQ**. Answers are highlighted. Below the paraphrased context are the results of each reader. *GPT: X -> O* indicates that GPT reader got wrong with original context, but right with paraphrased context.

| Benchmark | Gold | Paraphrased |
|---|---|---|
| HotPotQA | **Question: The VMAQT-1 logo is a female spirit in Irish mythology who heralds the death of what?**

Title: VMAQT-1

Marine Tactical Electronic Warfare Training Squadron 1 (VMAQT-1) is a United States Marine Corps electronic warfare training squadron consisting of EA-6B Prowler jets. The squadron is based at Marine Corps Air Station Cherry Point, North Carolina and falls under the command of Marine Aircraft Group 14 (MAG-14) and the 2nd Marine Aircraft Wing (2nd MAW). The VMAQT-1 logo is the Banshee, an Irish mythological figure foretelling death. Its motto is "Tairngreacht Bas," Gaelic for "Death Foretold."

Title: Banshee

A banshee ( ; Modern Irish "bean sí", from Old Irish: "ban síde" , ] , "woman of the fairy mound" or "fairy woman") is a female spirit in Irish mythology who heralds the death of ==a family member==, usually by shrieking or keening. Her name is connected to the mythologically-important tumuli or "mounds" that dot the Irish countryside, which are known as síde (singular "síd") in Old Irish. | **GPT:** VMAQT-1 is a US Marine Corps training squadron consisting of EA-6B Prowler jets based in North Carolina. Their logo is the Banshee, a female spirit from Irish mythology who predicts death in a family by screaming or keening. The Banshee's name is associated with the mounds in the Irish countryside known as "síde." The squadron's motto is "Tairngreacht Bas," meaning "Death Foretold" in Gaelic.
*GPT: O -> X, Claude: X -> X*

**Claude:** The VMAQT-1 logo features a banshee, which is a female spirit in Irish mythology. According to Irish mythology, banshees shriek or keen to foretell the death of ==a family member==. The banshee's name comes from síde, the mythological mounds in the Irish countryside. Therefore, the banshee in the VMAQT-1 logo heralds the death of a family member.
*GPT: O -> O, Claude: X -> O* |

Table 6: **An example of a gold context and paraphrased contexts in HotPotQA.** Answers are highlighted. Below the paraphrased context are the results of each reader. *GPT: X -> O* indicates that GPT reader got wrong with original context, but right with paraphrased context.

| Benchmark | Gold | Paraphrased |
|---|---|---|
| StrategyQA | **Question: Can Lamborghini's fastest model win a race against a Porsche 911?**
**Answer: yes**

Title: Lamborghini Veneno

The Lamborghini Veneno (Spanish pronunciation:[be neno]) is a limited production high performance sportscar manufactured by Italian automobile manufacturer Lamborghini. (...) The Lamborghini Veneno has a top speed of 356 km/h (221 mph) and has a 0–97 km/h (0–60 mph) acceleration time of 2.8 seconds. The car has a braking distance of 30 m (98.0 ft) from 97–0 km/h (60–0 mph), and can produce 1.41 G while cornering.

Title: Porsche 911

Porsche 911 GT1 on display at the Porsche Museum The Porsche 911 GT1 is a race car that was developed in 1996 for the GT1 class in the 24 Hours of Le Mans. In order to qualify for GT racing, 25 road-going models were built to achieve type homologation. The engine in the GT1 is rated at 608 PS (447 kW; 600 hp) (544 PS (400 kW; 537 hp) for the road version) and accelerated from 0–97 km/h in 3.3 seconds. The top speed stood at 330 km/h (205 mph). (...) | **GPT:** The Lamborghini Veneno is a limited production high performance sports car created to commemorate Lamborghini's 50th anniversary. It is based on the Lamborghini Aventador and was first showcased at the 2013 Geneva Motor Show with a price of $4 million, making it one of the most expensive cars in the world. It has a top speed of 356 km/h, can accelerate from 0-97 km/h in 2.8 seconds, and can brake from 97-0 km/h in 30 meters while cornering at 1.41 Gs. The Porsche 911 GT1 is a race car developed in 1996 for the GT1 class in the 24 Hours of Le Mans. In order to qualify for GT racing, 25 road-going models were built to achieve type homologation. The engine produces 608 PS (600 hp) for the race version and 544 PS (537 hp) for the road version. It can accelerate from 0-97 km/h in 3.3 seconds and has a top speed of 330 km/h. (...)
*GPT: X -> X, Claude: X -> O*

**Claude:** Based on the background documents, it seems that Lamborghini Veneno would likely win a race against a Porsche 911. The Veneno has a higher top speed of 356 km/h compared to 330 km/h for the Porsche 911 GT1. The Veneno also has faster acceleration, able to go from 0 to 97 km/h in just 2.8 seconds compared to 3.3 seconds for the Porsche. Although the Porsche 911 GT1 was designed primarily as a race car and the Veneno was more of a limited production sportscar, the Veneno still has superior performance specifications. Given its faster speed, acceleration and handling, the Lamborghini Veneno model would probably win a race against a Porsche 911.
*GPT: X -> O, Claude: X -> O* |

Table 7: **An example of a gold context and paraphrased contexts in StrategyQA.** Below the paraphrased context are the results of each reader. *GPT: X -> O* indicates that GPT reader got wrong with original context, but right with paraphrased context.

| Benchmark | Gold | Paraphrased |
|---|---|---|
| QASC | **Question: What are pollens? (A) eukaryotes (B) unicellular organisms (C) building blocks of organisms (D) antigens that cause allergy (E) bees (F) microscopic organisms (G) nucleotides composing polymers (H) diploid Answer: (D)**

Any antigen that causes an allergy is called an allergen. Many of the common allergens are pollens and dusts.. | **GPT:** Allergens are substances that cause allergies, and they include common sources such as pollens and dusts. Pollens are not unicellular organisms or bees, but rather they are **building blocks of organisms** composed of nucleotides that form polymers. They are not diploid or microscopic, but they are known to be antigens that can trigger allergic reactions.

*GPT: O -> X, Claude: X -> X*

**Claude:** Here is a paraphrased summary of the background documents:

Pollens are microscopic cells that contain the male gametes of plants and facilitate reproduction. They are produced by the anthers of stamens in flowering plants. When pollen grains get dispersed by wind, water, or insects and come in contact with the stigma of a flower, they germinate and produce a pollen tube to transfer the sperm cells to the ovule.

Though pollens are important for plant
*GPT: O -> X, Claude: X -> X* |
| | **Question: Climate is generally described in terms of what? (A) sand (B) occurs over a wide range (C) forests (D) Global warming (E) rapid changes occur (F) local weather conditions (G) measure of motion (H) city life Answer: (F)**

Climate is generally described in terms of temperature and moisture.

Fire behavior is driven by local weather conditions such as winds, temperature and moisture. | **GPT:** The climate is typically defined by its temperature and moisture levels. The behavior of fires is influenced by local weather conditions, which may include wind, temperature, and moisture.
*GPT: X -> O, Claude: X -> O*

**Claude:** Based on the background documents, climate is generally described in terms of temperature and moisture or local weather conditions, not the other options.
*GPT: X -> O, Claude: X -> O* |

Table 8: **Examples of gold contexts and paraphrased contexts in QASC.** Below the paraphrased context are the results of each reader. *GPT: X -> O* indicates that GPT reader got wrong with original context but right with paraphrased context. The first example shows a failure case, where the distractor option misleads the paraphraser to generate irrelevant information (see the **red-colored** part). The second example, on the other hand, shows a success case.