# OpenReview forum: "Knowledge Corpus Error in Question Answering"
_EMNLP/2023/Conference — EMNLP 2023 Findings_

### Official Review · Reviewer_JpPN · 2023-08-01

**Soundness:** 2

**Excitement:**

3: Ambivalent: It has merits (e.g., it reports state-of-the-art results, the idea is nice), but there are key weaknesses (e.g., it describes incremental work), and it can significantly benefit from another round of revision. However, I won't object to accepting it if my co-reviewers champion it.

**Paper Topic And Main Contributions:**

The work examines the phenomena of why generated passages, instead of retrieved passages from a corpus, can be more effective in the conventional definition of QA tasks. The work sets a target to find out why this gap occurs.

The gap: the index does not cover all the available knowledge, more relevant data/passages can exist outside of the index space. LLMs can look at the whole information space thus not missing out relevant information. The authors coin the term knowledge corpus error which defines a subspace of the knowledge space which the index of documents does not refer to, potentially missing relevant info when trying to answer a question in a QA task.

The paper includes one main focused benchmark the aims to evaluate the knowledge corpus error by paraphrasing golden passages of 4 different QA tasks, and evaluating the reading capabilities of 2 LLMs, and compared to setups using the actual gold passages representing the best possible retrieval function. The evaluation results show quite significant improvement in EM and accuracy is 3 out of the 4 tasks.

**Questions For The Authors:**

- Did you validate or have any conclusions on the generated paraphrases? are they different that what is included in the index?
- Why was is neccessary to use few-shot in NQ experiments?

**Reasons To Accept:**

- The paper focuses on an important understanding gap of QA systems and LLMs which has not been addressed yet.
- Evaluation results showing non-corpus memory can lead to increase in model performance, tested positively on 3 out of 4 QA datasets and 2 LLMs.
- Setup of experiments are detailed well for reproduction and provided in the appendix.
- Writing style, explanations and focused contribution.

**Reasons To Reject:**

* The main experiment suffers from a major drawback that it is unclear from the setup of the evaluations how well paraphrasing using a LLM generates good representations, that is, good examples of _external knowledge_ outside of the corpus. The method in the paper does not validate, or convince, that the paraphrases are indeed outside of the corpus knowledge.
* The paper's contribution is limited. The results of the evaluations just verify the known phenomena that is also addressed by the authors related work, and it quite intuitive to understand since Wikipedia is far smaller than the data large LLMs are trained on.

**Reproducibility:**

4: Could mostly reproduce the results, but there may be some variation because of sample variance or minor variations in their interpretation of the protocol or method.

**Reviewer Confidence:**

4: Quite sure. I tried to check the important points carefully. It's unlikely, though conceivable, that I missed something that should affect my ratings.

---

> ### Author Rebuttal · Authors · 2023-08-29
>
> Thanks for your thorough review and valuable feedback. We here go through three main points -- analysis on paraphrased passages, contribution of this paper, and use of few-shot in NQ.
>
> ## 1. Analysis on paraphrased passages
>
> > The main experiment suffers from a major drawback that it is unclear from the setup of the evaluations how well paraphrasing using a LLM generates good representations, that is, good examples of external knowledge outside of the corpus. The method in the paper does not validate, or convince, that the paraphrases are indeed outside of the corpus knowledge. (Reasons To Reject 1)
>
> > Did you validate or have any conclusions on the generated paraphrases? are they different that what is included in the index? (Question 1)
>
> We first want to clarify that we are **not** arguing that paraphrasing improves the reader performance as a result of incorporating "*external knowledge* outside of the corpus".
>
> As the name ***paraphrase*** and the prompts (Table 2 from the paper) suggest, we are not trying to generate new knowledge by LLMs. The role of paraphrasing is to smooth the textual information from the gold passage (= best-case scenario in retrieval). This would likely result in more factual and helpful passages compared to generation from scratch, like GenRead (Yu et al., 2023).
>
> ### Qualitative analysis on the source of knowledge corpus error
> In §3.3, we suggested two factors as the source of knowledge corpus error (= why paraphrasing improves the reader performance). Through further analysis, we identified three factors contributing to knowledge corpus error and have provided illustrative examples for each.
>
>
> ---
> 1. **More focus on the question**
> Gold passages are a fraction of Wikipedia, where the written communicative intent is to generally inform about the subject. Therefore, gold passages inevitably include irrelevant information to the question.
> In fact, it has been shown that “when both retrieved and generated documents contain the correct answer, the FiD reader can produce more correct answers when reading the generated documents” (Yu et al., 2023 - §4.4). It suggests that retrieved passages are more likely to contain noisy information that is irrelevant to the question.
> Below is our example from NQ. The gold passage focuses on general information about the TV series, while paraphrased ones has shifted their focus to the character the actor played, which is directly tied to the question.
>
>
> *Question/Answer*: who plays charles on have and have nots  / Nick Sager
>
> *Gold*:
> Title: The Haves and the Have Nots (TV series)
> The Road to Hell", it was revealed that David had a pulse while Erica's fate was a casualty and died from the explosion. Her death was the fuel for David’s warpath of vengeance on Veronica. - **Charles Frederickson (Nick Sager)**: (Season 4–present) The president-elect and the love interest to Candace, whom she uses to help her get her money back. (...)
>
> *GPT-paraphrased*:
> crew of "The Haves and the Have Nots": In the TV series, "The Haves and the Have Nots", **Charles Frederickson is played by Nick Sager**. He plays the character of the president-elect and Candace's love interest. (...)
>
> *Claude-paraphrased*:
> **Charles, played by Nick Sager**, is one of the main characters on the show The Haves and the Have Nots. He is portrayed as the president-elect and love interest of the character Candace. (...)
>
> ---
> 2. **Chain-of-thought**
> Some questions require composition of facts to answer the question. In such case, we saw that paraphrasing is doing something similar to chain-of-thought.
> Below is an example from HotPotQA (Table 4 in the paper). Paraphrased passage is combining two distinct facts into a single fact.
>
> *Question/Answer*: The VMAQT-1 logo is a female spirit in Irish mythology who heralds the death of what?  /  a family member
>
> *Gold*:
> Title: VMAQT-1
> (...) The VMAQT-1 logo is the Banshee, an Irish mythological figure foretelling death. Its motto is “Tairngreacht Bas,” Gaelic for “Death Foretold.”
> Title: Banshee
> A banshee ( ; Modern Irish "bean sí", from Old Irish: "ban síde" , ] , "woman of the fairy mound" or "fairy woman") is a female spirit in Irish mythology who heralds the death of a family member, usually by shrieking or keening. (...)
>
> *Claude-paraphrased*:
> The VMAQT-1 logo features a banshee, which is a female spirit in Irish mythology. According to Irish mythology, banshees shriek or keen to foretell the death of a family member. (...)
>
> ---
> 3. **Commonsense knowledge**
> Commonsense knowledge plays a crucial role in understanding the world, but not often explicitly stated (especially in Wikipedia). During paraphrasing, commonsense knowledge is expressed in the passage, which aids the reader to answer better.
> Below is an example from StrategyQA. *Someone who dropped out of college will probably not attend graduation ceremony* is an important commonsense knowledge here.
>
> *Question/Answer*: Was Bruce Lee absent from the 1964 University of Washington graduation ceremony? / True
>
> *Gold*:
> Title: Bruce Lee
> Lee is best known as a martial artist, but he also studied drama and Asian and Western philosophy while a student at the University of Washington and throughout his life. (...)
> Title: Bruce Lee
> Lee dropped out of college in early 1964 and moved to Oakland to live with James Yimm Lee. (...)
>
> *Claude-paraphrased*:
> Based on the background documents, it appears that Bruce Lee was likely absent from the 1964 University of Washington graduation ceremony. There is no mention of him attending or participating in the graduation in either document. The documents instead mention that in early 1964, Lee dropped out of college and moved to Oakland, California. (...) While this evidence is not definitive, it suggests that Lee probably did not attend or participate in the University of Washington's 1964 graduation ceremony.
>
> ---
> ## 2. Limited contribution
> > The paper's contribution is limited. The results of the evaluations just verify the known phenomena that is also addressed by the authors related work, and it quite intuitive to understand since Wikipedia is far smaller than the data large LLMs are trained on. (Reasons To Reject 2)
>
> We will first discuss our contribution in general, and then address specific limitations mentioned.
>
> Even though the phenomenon is intuitive to understand and presumable, **we believe our formulation and its follow-up experiment setup are novel**. Our analytic formulation (§3 in the paper) connects the use of generated passage to classic formulation of *retrieve-then-read* in QA. This helps understanding why generated passages can be more beneficial than retrieved passages (Yu et al., 2023). Further, it can serve as a cornerstone to understand the role LLM-generated text plays, compared to pre-existing text. As we mentioned in §2.1, this discourse is becoming increasingly important, as many LMs, especially the smaller open-source ones, are being trained on LLM-generated text.
>
> We notice the focus on our evaluations in the comment. While evaluation is an important part, we want to emphasize that those are rather an empirical follow-up to our formulation. We believe future works can utilize it to build further contributions.
>
> > The results of the evaluations just verify the known phenomena that is also addressed by the authors related work
>
> Although it is well-understood that LLM-generated text can benefit a lot of tasks including QA, there has been no controlled experiment to measure the limits of existing text (knowledge corpus). Analysis closest to ours is the “readability analysis” from §4.4 in Yu et al., 2023 (more details in "Qualitative analysis - 1" from above). However, not all passages containing the answer are guaranteed good, leaving a chance that the gap in reader performance is due to a suboptimal retrieval. In contrast, our experimental setup is focused to observe the inherent limit of the knowledge corpus, disregarding the performance of retrieval.
>
> > it quite intuitive to understand since Wikipedia is far smaller than the data large LLMs are trained on
>
> We believe that this phenomenon is not mainly about the size of the knowledge corpus vs the size of the data LM is trained on. Instead, we’d like to emphasize the inherent limit any fixed corpus would hold and language models’ generalization capability to output meaningful text beyond it. For further analysis, it will be interesting to use larger non-parametric memory than Wikipedia.
>
> ## 3. Use of few-shot prompting for NQ
> > Why was is neccessary to use few-shot in NQ experiments?
>
> We used 3-shot prompting for NQ because EM in zero-shot Claude reader was too low to deduce meaningful results. Performance gap between zero-shot and few-shot is explained by the answer format -- only single entity are counted correct whereas LLMs may output a whole sentence. Note that for such reason, the use of few-shot in NQ is prevalent (Gudibande et al., 2023; Touvron et al., 2023). HotPotQA faces similar problem, but we concluded that EMs were sufficiently high in zero-shot.
>
> ### References
> Gudibande et al., 2023, [The False Promise of Imitating Proprietary LLMs](https://arxiv.org/abs/2305.15717).
> Touvron et al., 2023, [Llama 2: Open Foundation and Fine-Tuned Chat Models](https://arxiv.org/abs/2307.09288).
> Yu et al., 2023, [Generate rather than Retrieve: Large Language Models are Strong Context Generators](https://openreview.net/forum?id=fB0hRu9GZUS), ICLR.

---

### Official Review · Reviewer_dTRR · 2023-08-03

**Typos Grammar Style And Presentation Improvements:** 1. I think the "not in Z" is not need…
**Soundness:** 3

**Excitement:**

4: Strong: This paper deepens the understanding of some phenomenon or lowers the barriers to an existing research direction.

**Paper Topic And Main Contributions:**

The paper introduces and formally describes “Knowledge Corpus Error”, which happens when the knowledge corpus for retrieval excludes more relevant content. The authors argue that knowledge corpus errors exist in three out of four selected datasets by showing reader models could achieve higher accuracy / exact match score with paraphrased documents.

**Questions For The Authors:**

Question A: Is it possible to distinguish between “good documents, bad reader” and “good reader, bad documents”? It seems you define knowledge corpus errors as the latter, but how could you know if the readers are good enough to extract information from the documents? Or how do you know if you are prompting the reader properly?
Question B: Per “Reason to reject” 1, how are the hyperparameters decided (e.g. Max token, temperature, etc.)?
Question C: In the definition of knowledge corpus error (L184), it is mentioned that knowledge corpus error happens when p(a|q, c˜) > p(a|q, c). However, when measuring the errors, the exact match metric is used instead of probability. Is there a reason for this, or could the authors show that the two are equivalent?
Question D: Is there a reason why an example for QASC is not provided?
Question E: Why is the training split for StrategyQA used, and how is the subset of HotPotQA selected?


**Reasons To Accept:**

1. “Knowledge Corpus Error” is a novel concept that can explain current LM errors and why generated context might be superior to retrieved ones in knowledge-intensive tasks. This work could inspire future work in locating errors coming from corpus that are not good enough.
2. The paper empirically shows that knowledge corpus errors exist in some of the current open-domain question answering settings.
3. The mathematical formulation of knowledge corpus is clear, easy to follow and mostly correct.


**Reasons To Reject:**

There are four points in total, ranked from major to minor:
1. The experiments are conducted only once, without generating multiple samples or using different hyperparameters. The reason for choosing the hyperparameters are not mentioned.
2. Lacking detailed analysis:
The authors do show that knowledge corpus errors could exist in some settings. However, they never discuss the reasons behind them. (e.g. How are the paraphrased documents different from the gold documents?) Also, the reasons they provide for the accuracy degradation in QASC are weak, and the authors did not provide any experiments to back them up. Seed facts are not necessarily better than Wikipedia documents, and no examples are shown to prove the paraphrased contexts are relevant to the distractors.
3. Unsupported Claims:
L308-L311 “We further track down the source of knowledge corpus error and suggest that reasoning and commonsense required in the benchmarks are closely tied to knowledge corpus error.”
However, the paper’s second limitation (as mentioned in the Limitation section) is “that the exact source of knowledge corpus error is unclarified”. I would suggest editing the last few sentences in the conclusion.
4. Results on more LMs could be included. Is this phenomenon consistent for all kinds of LMs, including ones that are not instruction-tuned or not trained with RLHF? What about LMs of different sizes?


**Reproducibility:**

4: Could mostly reproduce the results, but there may be some variation because of sample variance or minor variations in their interpretation of the protocol or method.

**Reviewer Confidence:**

4: Quite sure. I tried to check the important points carefully. It's unlikely, though conceivable, that I missed something that should affect my ratings.

---

> ### Author Rebuttal · Authors · 2023-08-29
>
> Thank you for your valuable feedback. We will first respond to the three main points from “Reasons To Reject”, then answer the queries.
>
> ## From "Reasons To Reject"
>
> > 1. The experiments are conducted only once, without generating multiple samples or using different hyperparameters. The reason for choosing the hyperparameters are not mentioned.
>
> Thank you for your observation and we apologize for the confusion. We respond to this as part of our answer to Question B. However, we would like to reiterate that the empirical findings are a secondary contribution of the paper after the analytical framework and introduction of the knowledge corpus error framework.
>
> > 2. Lacking detailed analysis: The authors do show that knowledge corpus errors could exist in some settings. However, they never discuss the reasons behind them. (e.g. How are the paraphrased documents different from the gold documents?) Also, the reasons they provide for the accuracy degradation in QASC are weak, and the authors did not provide any experiments to back them up. Seed facts are not necessarily better than Wikipedia documents, and no examples are shown to prove the paraphrased contexts are relevant to the distractors.
>
> ### Qualitative analysis on the source of knowledge corpus error
>
> In §3.3, we posited two factors as the source of knowledge corpus error (= why paraphrasing improves the reader performance). Through further analysis, we identified three factors contributing to knowledge corpus error and have provided illustrative examples for each.
>
> 1. **More focus on the question**
> Gold passages are a fraction of Wikipedia, where the written communicative intent is to generally inform about the subject. Therefore, gold passages inevitably include irrelevant information to the question.
> In fact, it has been shown that “when both retrieved and generated documents contain the correct answer, the FiD reader can produce more correct answers when reading the generated documents” (Yu et al., 2023 - §4.4). It suggests that retrieved passages are more likely to contain noisy information that is irrelevant to the question.
> Below is our example from NQ. The gold passage focuses on general information about the TV series, while paraphrased ones has shifted their focus to the character the actor played, which is directly tied to the question.
>
>
> *Question/Answer*: who plays charles on have and have nots  / Nick Sager
>
> *Gold*:
> Title: The Haves and the Have Nots (TV series)
> The Road to Hell", it was revealed that David had a pulse while Erica's fate was a casualty and died from the explosion. Her death was the fuel for David’s warpath of vengeance on Veronica. - **Charles Frederickson (Nick Sager)**: (Season 4–present) The president-elect and the love interest to Candace, whom she uses to help her get her money back. (...)
>
> *GPT-paraphrased*:
> crew of "The Haves and the Have Nots": In the TV series, "The Haves and the Have Nots", **Charles Frederickson is played by Nick Sager**. He plays the character of the president-elect and Candace's love interest. (...)
>
> *Claude-paraphrased*:
> **Charles, played by Nick Sager**, is one of the main characters on the show The Haves and the Have Nots. He is portrayed as the president-elect and love interest of the character Candace. (...)
>
> ---
> 2. **Chain-of-thought**
> Some questions require composition of facts to answer the question. In such case, we saw that paraphrasing is doing something similar to chain-of-thought.
> Below is an example from HotPotQA (Table 4 in the paper). Paraphrased passage is combining two distinct facts into a single fact.
>
> *Question/Answer*: The VMAQT-1 logo is a female spirit in Irish mythology who heralds the death of what?  /  a family member
>
> *Gold*:
> Title: VMAQT-1
> (...) The VMAQT-1 logo is the Banshee, an Irish mythological figure foretelling death. Its motto is “Tairngreacht Bas,” Gaelic for “Death Foretold.”
> Title: Banshee
> A banshee ( ; Modern Irish "bean sí", from Old Irish: "ban síde" , ] , "woman of the fairy mound" or "fairy woman") is a female spirit in Irish mythology who heralds the death of a family member, usually by shrieking or keening. (...)
>
> *Claude-paraphrased*:
> The VMAQT-1 logo features a banshee, which is a female spirit in Irish mythology. According to Irish mythology, banshees shriek or keen to foretell the death of a family member. (...)
>
> ---
> 3. **Commonsense knowledge**
> Commonsense knowledge plays a crucial role in understanding the world, but not often explicitly stated (especially in Wikipedia). During paraphrasing, commonsense knowledge is expressed in the passage, which aids the reader to answer better.
> Below is an example from StrategyQA. *Someone who dropped out of college will probably not attend graduation ceremony* is an important commonsense knowledge here.
>
> *Question/Answer*: Was Bruce Lee absent from the 1964 University of Washington graduation ceremony? / True
>
> *Gold*:
> Title: Bruce Lee
> Lee is best known as a martial artist, but he also studied drama and Asian and Western philosophy while a student at the University of Washington and throughout his life. (...)
> Title: Bruce Lee
> Lee dropped out of college in early 1964 and moved to Oakland to live with James Yimm Lee. (...)
>
> *Claude-paraphrased*:
> Based on the background documents, it appears that Bruce Lee was likely absent from the 1964 University of Washington graduation ceremony. There is no mention of him attending or participating in the graduation in either document. The documents instead mention that in early 1964, Lee dropped out of college and moved to Oakland, California. (...) While this evidence is not definitive, it suggests that Lee probably did not attend or participate in the University of Washington's 1964 graduation ceremony.
>
> ---
>
> ### Analysis on QASC
>
> We apologize for the lack of clarity in our explanation for performance degradation for QASC (§4.3). Here we provide examples to better illustrate our explanation. We will include further details in the final version of the manuscript.
>
>
> 1. **Gold passages we used are manually created, instead of being extracted from raw Wikipedia.**
> Gold passages we used represent the best-case scenario in retrieval, thereby eliminating any retrieval error. However, gold passages we used for QASC do not reflect this setting, as they are **manually selected** from cleaned knowledge sources like WorldTree corpus. Such choice was done because there were no annotated supporting passage from raw source such as Wikipedia. An example below illustrates how manually created "gold" in QASC differs from other benchmarks.
>
>
> *QASC*:  Any antigen that causes an allergy is called an allergen. Many of the common allergens are pollens and dusts.
> *NQ*: Title: Boom (P.O.D. song)
> Boom (P.O.D. song) "Boom" is a song by American rock band P.O.D. It was released in May 2002 as the third single from their second major label studio album "Satellite". While it did not chart as well as the album's previous singles, the song has appeared significantly in film and television. "Boom (The Crystal Method remix)" was included on the remix album "Community Service" and as a bonus track on the special edition re-release of "Satellite" available August 27
>
> ---
>
>
> 2. **Distractor options in multiple-choice question confuses the model to generate passage relevant to those options.**
> This is best illustrated through an example. See the bold parts to see how distractor option (C) misled the paraphraser, GPT.
>
> *Question*: What are pollens? (A) eukaryotes (B) unicellular organisms **(C) building blocks of organisms** (D) antigens that cause allergy (E) bees (F) microscopic organisms (G) nucleotides composing polymers (H) diploid
>
> *Answer*: D
>
> *Gold passage*: Any antigen that causes an allergy is called an allergen. Many of the common allergens are pollens and dusts.
>
> *GPT-paraphrased passage*: Allergens are substances that cause allergies, and they include common sources such as pollens and dusts.  Pollens are not unicellular organisms or bees, but rather **they are building blocks of organisms** composed of nucleotides that form polymers. They are not diploid or microscopic, but they are known to be antigens that can trigger allergic reactions.
>
> ---
>
> > 3. Unsupported Claims: L308-L311
>
> Thank you for pointing this out.  This stems from the analytical framework in Section 3.  We did include a statement in the limitations section L323-L327 adding a caveat. However,  we can see how it would be unclear.  While we have introduced an analytical framework backed up by empirical results, there are many factors that contribute to this error. We will remove this claim from the final version of the paper, focusing on our concrete contributions instead.
>
>
> > 4. Results on more LMs could be included. Is this phenomenon consistent for all kinds of LMs, including ones that are not instruction-tuned or not trained with RLHF? What about LMs of different sizes?
>
> First of all, we want to distinguish the use of LMs for *paraphrasing* and *reading* (answering questions). LMs used for *paraphrasing* are the means of observation, but LMs used for *reading* are the subjects of observation. The purpose of paraphrasing the gold passage is to obtain passage even better than gold. Therefore, we believe that only using the best LLMs available for paraphrasing is justified.
>
> However, we definitely notice the benefit of including more models for readers. This will make our argument more robust. We are especially eyeing on the recent open-source LMs such as Llama-2 and Falcon. We are looking forward to add more LMs, both base and instruction-tuning+RLHFed, in the future version.
>
> We'd like to further mention that the reader can be a supervised model trained on QA datasets, e.g., FiD (Izacard and Grave, 2021). From our experiment, FiD trained on retrieved passages, $c \in \mathcal{Z}$, performed worse when given paraphrased passages, $c \notin \mathcal{Z}$ ($\mathcal{Z}$ refers to knowledge corpus such as Wikipedia). This indicates a distribution shift after paraphrasing. Training FiD on paraphrased passages would be required to see a true effect of paraphrasing for FiD.
>
>
> ## From "Questions For The Authors"
> ### Question A
> > Is it possible to distinguish between “good documents, bad reader” and “good reader, bad documents”? It seems you define knowledge corpus errors as the latter, but how could you know if the readers are good enough to extract information from the documents? Or how do you know if you are prompting the reader properly?
>
> *Good is in the eye of the reader*. We think of *good* documents as documents beneficial to the reader, i.e., a reader-dependent concept.
> $$
> p(a|q) = \sum p(a|q,c)p(c|q)
> $$
> From above, we can think of the probability on the right-hand side being modeled by different models. It is possible for some $c$, $p_{1}(a|q,c) \neq p_{2}(a|q,c)$. The case of using different prompts with the same model can be thought equivalent to using different models.
>
> However, as we are not conditioning on the reader model when paraphrasing the context, we are assuming some level of **universality of *good* documents** across various readers. Although this universality is presumable, it will be worth exploring the difference between *good* documents depending on the readers, from LLMs and supervised FiD to human.
>
> In fact, we found out that there seems to be some universality of *goodness* across the readers we used. Table 1 shows exact match in HotPotQA for two readers (GPT-3.5 and Claude) . For majority of the examples, two readers accord with each other (both correct or wrong), and this ratio is even higher in paraphraed contexts (70% for gold, 77% for GPT-paraphrased, 81% for Claude-paraphrased).
>
> | Who is correct? (%)   | Both | Only GPT | Only Claude | None (both wrong) |
> |--------------------|------|----------|-------------|-------------------|
> | Gold               | 27   | 9        | 21          | 43                |
> | GPT-paraphrased    | 33   | 5        | 18          | 44                |
> | Claude-paraphrased | 39   | 4        | 15          | 42                |
> <Table 1: Exact Match in HotPotQA>
>
> > Question B: Per “Reason to reject” 1, how are the hyperparameters decided (e.g. Max token, temperature, etc.)?
>
> Generation hyperparameters (max tokens, temperature, zero-shot vs few-shot) are decided through trial-and-error. For example, max tokens was adjusted according to few preliminary samples. We tried to tweak temperature for QASC after observing deviant result, but only had minor impact. 3-shot setup was chosen for NQ because the performance was too low in zero-shot.
>
> > Question C: In the definition of knowledge corpus error (L184), it is mentioned that knowledge corpus error happens when p(a|q, c˜) > p(a|q, c). However, when measuring the errors, the exact match metric is used instead of probability. Is there a reason for this, or could the authors show that the two are equivalent?
>
> Thanks for pointing out an important gap. We used EM / accuracy for two reasons.
>
> 1. The reader models we used, GPT-3 and Claude, are black-box APIs where probability cannot be directly known. Therefore, we used end metrics (exact match / accuracy) as proxies.
>
> 2. In experiments, end metrics (exact match / accuracy) are far more intuitive than probability. This helps the reader to grasp the effect of paraphrasing. In addition, it allows comparison with other works as these are the most widely reported figures.
>
> We acknowledge the gap between our analytic formulation (probability) and empirical experiments (exact match / accuracy). However, we believe that this gap is negligible, especially given that we adopt greedy decoding. Furthermore, we believe the aforementioned benefits outweigh this limitation.
>
> > Question D: Is there a reason why an example for QASC is not provided?
>
> We apologize for our negligence. In addition to an example provided in #2 above, we here provide an example where paraphrasing was successful.
>
> ---
> *Question*: Climate is generally described in terms of what? (A) sand (B) occurs over a wide range (C) forests (D) Global warming (E) rapid changes occur (F) local weather conditions (G) measure of motion (H) city life
> *Answer*: F
>
> *Gold*: Climate is generally described in terms of temperature and moisture.\n\nFire behavior is driven by local weather conditions such as winds, temperature and moisture.
> *GPT-paraphrased*: The climate is typically defined by its temperature and moisture levels. The behavior of fires is influenced by local weather conditions, which may include wind, temperature, and moisture.
> *Claude-paraphrased*: Based on the background documents, climate is generally described in terms of temperature and moisture or local weather conditions, not the other options.
>
> *GPT reader given gold / GPT-paraph / Claude-paraph*: X / O / O
> *Claude reader given gold / GPT-paraph / Claude-paraph*: X / O / O
>
> ---
> > Question E: Why is the training split for StrategyQA used, and how is the subset of HotPotQA selected?
>
> We used the training split for StrategyQA because the dev split contained too few (490) examples. For HotPotQA, we used the first 1531 samples in the dataset.
>
> ## References
> Izacard and Grave, 2021, [Leveraging Passage Retrieval with Generative Models
> for Open Domain Question Answering](https://aclanthology.org/2021.eacl-main.74/), EACL.

---

### Official Review · Reviewer_bwTs · 2023-08-04

**Soundness:** 2

**Excitement:**

2: Mediocre: This paper makes marginal contributions (vs non-contemporaneous work), so I would rather not see it in the conference.

**Missing References:**

1. Alex Mallen, Akari Asai, Victor Zhong, Rajarshi Das, Daniel Khashabi, Hannaneh Hajishirzi. When Not to Trust Language Models: Investigating Effectiveness of Parametric and Non-Parametric Memories. The Annual Conference of the Association for Computational Linguistics (ACL), 2023.

**Paper Topic And Main Contributions:**

This paper introduces a new prompting method for performing open-retrieval QA with LLMs. In their method, after performing retrieval the authors use the LLM to first paraphrasing the retrieved passage before using the summarized passage in a separate prompt to perform QA. The authors demonstrate that performing this prompting method improves QA performance across a variety of LLMS (Claude + GPT3) and QA datasets.

**Reasons To Accept:**

The proposed method is simple, intuitive and easy to implement. It also shows strong performance gains in a variety of settings.

**Reasons To Reject:**

The authors suggest several explanations for the poor performance on QASC. Further analysis is required to identify and understand why this method fails in these settings.

While the proposed method improves in a variety of settings, it's unclear why this is the case. The authors suggest that paraphrasing allows the model to incorporate knowledge that lies outside the knowledge corpus, but there is no analysis justifying this fact. Other possible explanations are that the model in summarizing the passage given the question is (1) doing something akin to chain-of-though prompting or (2) is simply rephrasing the passage into a more usable format, without introducing any additional information. Furthermore, it's unclear whether (3) "knowledge corpus error", the suggested reason for this improved performance, is a desirable feature as it can also lead to hallucination. Further analysis is required to determine the source of these gains and justify the author's proposed reasoning.

Another type of analysis that would strengthen this work is understanding what types of questions this method works for. [1 from missing references] demonstrates that retrieval hurts performance for questions about popular entities. It is very possible that the proposed method is simply mitigating the negative effects of retrieval in such instances, and eliminating retrieval in these instances would achieve a similar effect.

**Reproducibility:**

3: Could reproduce the results with some difficulty. The settings of parameters are underspecified or subjectively determined; the training/evaluation data are not widely available.

**Reviewer Confidence:**

4: Quite sure. I tried to check the important points carefully. It's unlikely, though conceivable, that I missed something that should affect my ratings.

---

> ### Author Rebuttal · Authors · 2023-08-29
>
> Thank you for your valuable feedback. We will first provide a general comment regarding the reasons to accept, before discussing responses to the three main points from “Reasons To Reject” and then close with an additional analysis that may find useful.
>
>
> # Reasons to accept
>
>
> Thank you for identifying the ease of implementation of our approach. We hope that it can be used to aid reproducibility for other researchers. However, we wish to assert that our contributions are more than a “simple method to gain performance advantages”.
>
>
> What we pose is a new framework to help researchers reason about how knowledge is incorporated in open-domain NLP tasks combining retrieval and reasoning. We hope that the empirical findings of our paper help justify our argument. However, our principle argument centers around the source of the error. While we show that the LM can aid with smoothing the errors with pipeline, we wish to remind the reviewer that our contributions also include the analytical framework to reason with these errors.
>
>
> # Reasons to reject
> ## Analysis on QASC
>
>
> We apologize for the lack of clarity in our explanation for performance degradation for QASC (§4.3). Here we provide examples to better illustrate our explanation. We will include further details in the final version of the manuscript.
>
>
> 1. **Gold passages we used are manually created, instead of being extracted from raw Wikipedia.**
> Gold passages we used represent the best-case scenario in retrieval, thereby eliminating any retrieval error. However, gold passages we used for QASC do not reflect this setting, as they are **manually selected** from cleaned knowledge sources like WorldTree corpus. Such choice was done because there were no annotated supporting passage from raw source such as Wikipedia. An example below illustrates how manually created "gold" in QASC differs from other benchmarks.
>
>
>
>
> *QASC*:  Any antigen that causes an allergy is called an allergen. Many of the common allergens are pollens and dusts.
> *NQ*:
> Title: Boom (P.O.D. song)
> Boom (P.O.D. song) "Boom" is a song by American rock band P.O.D. It was released in May 2002 as the third single from their second major label studio album "Satellite". While it did not chart as well as the album's previous singles, the song has appeared significantly in film and television. "Boom (The Crystal Method remix)" was included on the remix album "Community Service" and as a bonus track on the special edition re-release of "Satellite" available August 27
>
>
> ---
>
>
>
>
> 2. **Distractor options in multiple-choice question confuses the model to generate passage relevant to those options.**
> This is best illustrated through an example. See the bold parts to see how distractor option (C) misled the paraphraser, GPT.
>
> *Question*: What are pollens? (A) eukaryotes (B) unicellular organisms **(C) building blocks of organisms** (D) antigens that cause allergy (E) bees (F) microscopic organisms (G) nucleotides composing polymers (H) diploid
>
>
> *Answer*: D
>
>
> *Gold passage*: Any antigen that causes an allergy is called an allergen. Many of the common allergens are pollens and dusts.
>
>
> *GPT-paraphrased passage*: Allergens are substances that cause allergies, and they include common sources such as pollens and dusts.  Pollens are not unicellular organisms or bees, but rather **they are building blocks of organisms** composed of nucleotides that form polymers. They are not diploid or microscopic, but they are known to be antigens that can trigger allergic reactions.
>
>
> ---
>
>
> ## Details on our “paraphrase” experiment
> We believe our intent was not delivered well, so we will first reiterate our argument, which will help answer the questions.
> > "The authors suggest that paraphrasing allows the model to incorporate knowledge that lies outside the knowledge corpus"
>
>
>
>
> This is **not** our argument. Our argument on *knowledge corpus error* is rooted in the formulation rather than intuition. Knowledge corpus error arises when some context outside of knowledge corpus is ignored even though it is better than the context inside the knowledge corpus (=context being retrieved). Formally speaking, for question $q$ and knowledge corpus $\mathcal{Z}$, $p_{retriever}(\tilde{c} \notin \mathcal{Z}|q) = 0$ where $p(\tilde{c}|q) > p(c \in \mathcal{Z}|q)$.
>
> This concept by itself doesn't assert anything about the source of knowledge corpus error (= source of improvement from paraphrasing), leaving a room for further works to contribute explanations specific to model types or tasks. From our manual inspection, we concluded three main factors contributing to knowledge corpus error (developed from §3.3).
> 1. More focus on the question
> 2. Chain-of-thought
> 3. Commonsense knowledge
>
>
> For conciseness, we defer detailed explanation to the very last section and answer the questions.
>
>
>
>
> > (1) "doing something akin to chain-of-thought prompting"
>
>
> Chain-of-thought definitely explains the benefit from paraphrasing. In fact, successful composition of multiple facts is often observed in cases where paraphrasing benefits the reader.
>
>
> > (2) "simply rephrasing the passage into a more usable format"
>
>
> This is exactly our point, hence we named our experiment *paraphrase*.
>
>
> > (3) "can also lead to hallucination"
>
>
> Hallucination is definitely a downside of using passages outside of knowledge corpus, as they usually require generation from LMs. Nonetheless, we’d like to emphasize that paraphrasing results in less hallucination compared to generation from scratch such as GenRead (Yu et al., 2023). Still, it will be important to find out whether LMs are faithfully paraphrasing the given gold passage, not adding any information (no matter it is factually correct or not).We will address this question in the future study.
>
>
> ## Instances where retrieval hurts / Comparison with pure LM generation
> First, we want to be clear that we did not actually retrieved the passages. We rather **used gold passages as the best-case scenario in retrieval**. Mallen et al., 2023 has shown that “non-parametric memories can mislead LMs”. However, they explain this with low recall@1 of retrieved passages for the questions the reader failed. In contrast, for the passages we used, recall@1 is 100%, because they are the gold which contain the answer.
>
>
> > Another type of analysis that would strengthen this work is understanding what types of questions this method works for.
>
>
> This analysis will be helpful to understand why knowledge corpus error arises. In fact, we suggested a short comparison between benchmarks in the second paragraph of §4.3. We found out that the benefit of paraphrasing is greater in HotPotQA(4.8) and StrategyQA(8.0) than NQ(3.1). This is probably because the former requires reasoning and commonsense, whereas NQ mostly requires simple fact recall.
> In addition to our benchmark-wise analysis, question-wise analysis may also be useful. We will consider it in the future version.
>
>
> > It is very possible that the proposed method is simply mitigating the negative effects of retrieval in such instances, and eliminating retrieval in these instances would achieve a similar effect.
>
>
> This is an interesting suggestion. There may be a chance that gold passaegs are harming the readers. If so, “non-parametric knowledge misleading LMs* will require further explanation that lower recall@1 in retrieved passages (Mallen et al., 2023), as recall@1 is 100% for gold passages. We look forward to delve into this question in further study.
>
>
> ## Appendix: Qualitative analysis on the source of knowledge corpus error
> In §3.3, we posited two factors as the source of knowledge corpus error (= why paraphrasing improves the reader performance). Through further analysis, we identified three factors contributing to knowledge corpus error and have provided illustrative examples for each.
>
> 1. **More focus on the question**
> Gold passages are a fraction of Wikipedia, where the written communicative intent is to generally inform about the subject. Therefore, gold passages inevitably include irrelevant information to the question.
> In fact, it has been shown that “when both retrieved and generated documents contain the correct answer, the FiD reader can produce more correct answers when reading the generated documents” (Yu et al., 2023 - §4.4). It suggests that retrieved passages are more likely to contain noisy information that is irrelevant to the question.
> Below is our example from NQ. The gold passage focuses on general information about the TV series, while paraphrased ones has shifted their focus to the character the actor played, which is directly tied to the question.
>
>
> *Question/Answer*: who plays charles on have and have nots  / Nick Sager
>
> *Gold*:
> Title: The Haves and the Have Nots (TV series)
> The Road to Hell", it was revealed that David had a pulse while Erica's fate was a casualty and died from the explosion. Her death was the fuel for David’s warpath of vengeance on Veronica. - **Charles Frederickson (Nick Sager)**: (Season 4–present) The president-elect and the love interest to Candace, whom she uses to help her get her money back. (...)
>
> *GPT-paraphrased*:
> crew of "The Haves and the Have Nots": In the TV series, "The Haves and the Have Nots", **Charles Frederickson is played by Nick Sager**. He plays the character of the president-elect and Candace's love interest. (...)
>
> *Claude-paraphrased*:
> **Charles, played by Nick Sager**, is one of the main characters on the show The Haves and the Have Nots. He is portrayed as the president-elect and love interest of the character Candace. (...)
>
> ---
> 2. **Chain-of-thought**
> Some questions require composition of facts to answer the question. In such case, we saw that paraphrasing is doing something similar to chain-of-thought.
> Below is an example from HotPotQA (Table 4 in the paper). Paraphrased passage is combining two distinct facts into a single fact.
>
> *Question/Answer*: The VMAQT-1 logo is a female spirit in Irish mythology who heralds the death of what?  /  a family member
>
> *Gold*:
> Title: VMAQT-1
> (...) The VMAQT-1 logo is the Banshee, an Irish mythological figure foretelling death. Its motto is “Tairngreacht Bas,” Gaelic for “Death Foretold.”
> Title: Banshee
> A banshee ( ; Modern Irish "bean sí", from Old Irish: "ban síde" , ] , "woman of the fairy mound" or "fairy woman") is a female spirit in Irish mythology who heralds the death of a family member, usually by shrieking or keening. (...)
>
> *Claude-paraphrased*:
> The VMAQT-1 logo features a banshee, which is a female spirit in Irish mythology. According to Irish mythology, banshees shriek or keen to foretell the death of a family member. (...)
>
> ---
> 3. **Commonsense knowledge**
> Commonsense knowledge plays a crucial role in understanding the world, but not often explicitly stated (especially in Wikipedia). During paraphrasing, commonsense knowledge is expressed in the passage, which aids the reader to answer better.
> Below is an example from StrategyQA. *Someone who dropped out of college will probably not attend graduation ceremony* is an important commonsense knowledge here.
>
> *Question/Answer*: Was Bruce Lee absent from the 1964 University of Washington graduation ceremony? / True
>
> *Gold*:
> Title: Bruce Lee
> Lee is best known as a martial artist, but he also studied drama and Asian and Western philosophy while a student at the University of Washington and throughout his life. (...)
> Title: Bruce Lee
> Lee dropped out of college in early 1964 and moved to Oakland to live with James Yimm Lee. (...)
>
> *Claude-paraphrased*:
> Based on the background documents, it appears that Bruce Lee was likely absent from the 1964 University of Washington graduation ceremony. There is no mention of him attending or participating in the graduation in either document. The documents instead mention that in early 1964, Lee dropped out of college and moved to Oakland, California. (...) While this evidence is not definitive, it suggests that Lee probably did not attend or participate in the University of Washington's 1964 graduation ceremony.
>
>
> ### References
> Yu et al., 2023, [Generate rather than Retrieve: Large Language Models are Strong Context Generators](https://openreview.net/forum?id=fB0hRu9GZUS), ICLR.

---

### Meta-Review · Area_Chair_rhbt · 2023-09-17

**Recommendation:** 3

**Metareview:**

This work provides notable improvements in performance by exploring an interesting question around “knowledge corpus error”. Reviewers expressed concerns about the impression that the method is augmenting “external knowledge” (presumably from your use of “passages outside the knowledge corpus”). We strongly recommend this wording is adjusted and clarified to bring the intended claims in line with the reviewers’ perceived claims. Additional analysis into why and how this works, as suggested by reviewer bwTs would also greatly strengthen this work. However, we do believe overall this set of experiments yield interesting and noteworthy results for a short paper.

---

### Decision · Program_Chairs · 2023-10-07

**Decision:**

Accept-Findings

**Comment:**

This work provides notable improvements in performance by exploring an interesting question around “knowledge corpus error”. Reviewers expressed concerns about the impression that the method is augmenting “external knowledge” (presumably from your use of “passages outside the knowledge corpus”). We strongly recommend this wording is adjusted and clarified to bring the intended claims in line with the reviewers’ perceived claims. Additional analysis into why and how this works, as suggested by reviewer bwTs would also greatly strengthen this work. However, we do believe overall this set of experiments yield interesting and noteworthy results for a short paper.